# Submodular Maximization via Gradient Ascent: The Case of Deep Submodular Functions

**Wenruo Bai[‡], William S Noble[*$], Jeff A. Bilmes[‡$]**
Depts. of Electrical & Computer Engineering[‡], Computer Science and Engineering[$], and Genome Sciences[*]
Seattle, WA 98195
{wrbai,wnoble,bilmes}@uw.edu

## Abstract

We study the problem of maximizing deep submodular functions (DSFs) [13, 3] subject to a matroid constraint. DSFs are an expressive class of submodular functions that include, as strict subfamilies, the facility location, weighted coverage, and sums of concave composed with modular functions. We use a strategy similar to the continuous greedy approach [6], but we show that the multilinear extension of any DSF has a natural and computationally attainable concave relaxation that we can optimize using gradient ascent. Our results show a guarantee of $\max_{0<\delta<1}(1-\epsilon-\delta-e^{-\delta^2\Omega(k)})$ with a running time of $O(n^2/\epsilon^2)$ plus time for pipage rounding [6] to recover a discrete solution, where $k$ is the rank of the matroid constraint. This bound is often better than the standard $1-1/e$ guarantee of the continuous greedy algorithm, but runs much faster. Our bound also holds even for fully curved ($c=1$) functions where the guarantee of $1-c/e$ degenerates to $1-1/e$ where $c$ is the curvature of $f$ [37]. We perform computational experiments that support our theoretical results.

## 1 Introduction

A set function $f : 2^V \to \mathbb{R}_+$ is called submodular [15] if $f(A) + f(B) \geq f(A \cup B) + f(A \cap B)$ for all $A, B \subseteq V$, where $V = [n]$ is the ground set. An equivalent definition of submodularity states that $f(v|A) \geq f(v|B)$ for all $A \subseteq B \subseteq V$ and $v \in V \setminus B$, where $f(v|A) \equiv f(\{v\} \cup A) - f(A)$ is the gain of element $v$ given $A$. This property of *diminishing returns* well models concepts such as information, diversity, and representativeness. Recent studies have shown that submodularity is natural for a large number of real world machine learning applications such as information gathering [23], probabilistic models [12], image segmentation [22], string alignment [28], document and speech summarization [27, 26], active learning [39], genomic assay selection [40] and protein subset selection [25], as well as many others.

In addition to having a variety of natural applications in machine learning, the optimization properties of submodular functions appear to be ever more auspicious. On one hand, the submodular minimization problem can be exactly solved in polynomial time [29, 11, 15]. Recent studies mostly focus on improving running times [24, 7]. On the other hand, submodular maximization is harder, and the optimal solution cannot be found by any polynomial time algorithm. A good approximate solution, however, is usually acceptable, and a simple greedy algorithm can find a constant factor $1 - 1/e$ approximate solution for the monotone non-decreasing[1] submodular maximization problem subject to a $k$-cardinality constraint [32]. Although submodular maximization is a purely combinatorial problem, there are also approaches to solve it via continuous relaxation (e.g. multilinear extension). For example, [6] offers a randomized continuous greedy algorithm that offers the same $1 - 1/e$ bound for monotone non-decreasing submodular maximization subject to a more general matroid

independence constraint. If the function's curvature $c$ is taken into account, this approach yields an improved guarantee of no worse than $1 - c/e$ [37]. Recent studies showed stochastic projected gradient methods [18, 31] can be useful on maximizing continuous DR-submodular function [35] [2]. The best guarantee is $(1 - OPT/e - \epsilon)$ by $1/\epsilon^3$ iterations in gradient methods [31].

The above results apply to any non-negative monotone submodular function. In practice, solving a given problem requires applying the algorithm to a specific submodular function, for example, set cover [16, 20], facility location [38], feature-based [21], graph cut [19] and deep submodular functions (DSF) [13, 3]. When working with a specific sub-class of functions, we can benefit from knowing the specific form and its mathematical properties. For example, in the simplest case, maximizing a modular function (i.e., a function $f$ for which both $f$ and $-f$ are submodular) under a matroid constraint can be exactly solved by a greedy algorithm. [20] showed benefit for submodular maximization in the specific case of weighted coverage functions.

In our work, we focus on DSF maximization under a matroid constraint. Introduced in [13, 3], DSFs are a generalization of set coverage, facility location, and feature-based functions. Importantly, the class of DSFs is a strict superset of the union of these three, which means that any method designed for a general DSF can be applied to set coverage, facility location, and feature-based functions but not vice versa. For example, $\sqrt{m(A)}$ is concave over modular; a feature-based function has the form of a sum of concave composed with modular functions, such as $\sqrt{m_1(A)} + \log(1 + m_2(A))$, while a two-layer DSF has a nested composition of the form $\sqrt{\sqrt{m_1(A)} + \arctan(m_2(A))} + \left[m_3(A) + \sqrt{m_4(A)}\right]^{1/4}$. In [3], it was shown that the expressivity of DSFs strictly grows with the number of layers.

To our knowledge, there have been no studies on the specific problem of DSF maximization. On the one hand, we can use the generic greedy or continuous greedy algorithms for DSF, since DSF is monotone submodular, but we should not be surprised if better bounds than $1 - 1/e$ can be achieved using the structure and properties of a DSF. The major contribution of the present work is to show that a very natural and computationally easy-to-obtain concave extension of DSFs is a nearly tight relaxation of the DSF's multilinear extension. Therefore, given this extension, we can use projected gradient ascent (Algorithm. 1 [4]) to maximize the concave extension and obtain a fractional solution, and then use pipage rounding [2, 6] to recover a discrete solution.

Our approach has the following advantages over the continuous greedy algorithm with only oracle access to the submodular function:

1. Easy concave extension: A natural concave extension of any DSF is easy to obtain, unlike the multilinear extension which often itself needs to be approximated using sampling.

2. Better guarantee for large $k$: Our method has a guarantee of $\max_{0 < \delta < 1}(1 - \epsilon - \delta - e^{-\delta^2 \Omega(k)})$, where $k$ is the rank of the matroid constraint (Corollary 2). A more complete formulation is $\max_{0 < \delta < 1}(1 - \epsilon)(1 - \delta)\left[1 - |V^{(1)}|e^{-\frac{\delta^2 w_{\min} k}{w_{\max}}}\right]$, where $w_{\min}/w_{\max}$ is the ratio of the smallest to the largest DSF element in the first weight layer of a DSF and $|V^{(1)}|$ is the size of the feature layer (see Figure 2). Importantly, this bound holds even when the curvature [37] of the DSF is $c = 1$ so the $1 - c/e$ bound of [37] is at its worst at $1 - 1/e$ (Lemma 7 in Appendix). We compare our bound with the traditional $1/2$ (for the greedy) and $1 - 1/e$ (for the continuous greedy) bounds in Figure 1. We show that our bound is better than the continuous greedy algorithm $(1 - 1/e)$ for large $k$ ($> 10^2 \sim 10^4$ depending on $k$ and $w_{\min}/w_{\max}$).

3. Improved running time: Other than the fact that a natural concave extension of a DSF is readily available, the running time of our method is $O(n^2 \epsilon^{-2})$ and is thus better than the $O(n^7)$ cost for the continuous greedy algorithm. Most of the continuous greedy algorithm's running time is for estimating the multilinear extension ($O(n^5)$ [6]), while in our method, calculating the DSF concave extension only needs one evaluation of the original function.

## 1.1 Background and Related work

[13, 3] introduced **deep submodular functions** where [3] discussed their theoretical properties and [13] their training in a fashion similar to how deep neural networks may be trained. Particularly relevant to the present study, [3] showed that while DSFs cannot express all submodular functions,

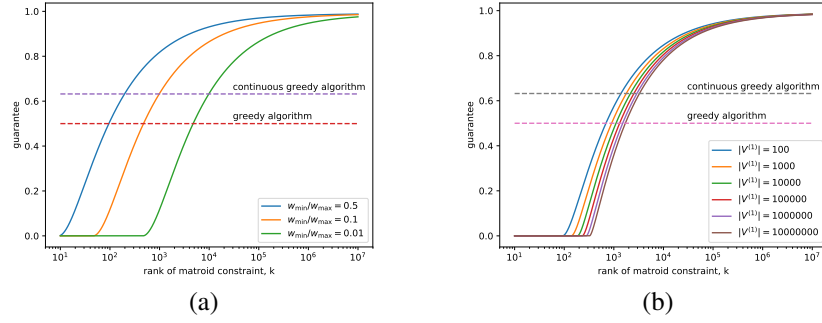

$$\text{(a)} \qquad\qquad \text{(b)}$$

Figure 1: Guarantee of propose methods stated in Theorem 3. Solid lines are the proposed guarantees with respect to the rank of matroid constraint; dash lines are guarantees for the continuous greedy algorithm and the greedy algorithm. Our guarantee is proportional to $1 - \epsilon$ and in the above figure, we use $\epsilon = 0.01$ for illustration. (a) is fixing $|V^{(1)}| = 10$ and each trace is for different $w_{\min}/w_{\max}$, which is the ratio of the smallest feature to the largest feature. (b) is fixing $w_{\min}/w_{\max} = 0.1$ and each trace is for different $|V^{(1)}|$, which is the size of the features layer (see Figure 2).

$k$-layer DSFs strictly generalize $k - 1$-layer DSFs. Moreover, the following classes of functions are all strict subclasses of DSFs [3].

1. Sums of concave composed with non-negative modular functions plus an arbitrary modular function (SCMM), also called feature-based functions [21], or "decomposable" submodular functions [36]. These functions take the form $f(A) = \sum_i \alpha_i \phi_i(m_i(A)) + m_\pm(A)$ where $\alpha_i$ are non-negative numbers, $\phi_i$ are monotone non-decreasing concave functions, $m_i$ are non-negative modular functions, and $m_\pm$ is an arbitrary modular function.
2. Weighted cardinality truncation (WCT) functions. $f(A) = \sum_i \alpha_i \min(|A \cap V_i|, \beta_i)$ where $V_i$ are subsets of $V$, $\alpha_i$ are non-negative numbers, and $\beta_i$ are non-negative integers.
3. Weighted coverage (WC) functions which take the form $f(A) = \sum_i \alpha_i \min(|A \cap V_i|, 1)$. See below.
4. Facility location (FL) functions. $f(A) = \sum_{i \in V} \max_{j \in A} w_{ij}$ where $w_{ij}$ is a matrix of non-negative numbers. It is a subclass of weighted coverage functions [20].

In particular, we have the following chain relationship between these classes of functions: FL $\subset$ WC $\subset$ WCT $\subset$ SCMM $\subset$ DSF $\subset$ All-Submodular-Functions [3]. In the present paper, we address any function that can be represented as a DSF.

In [20], submodular maximization of the special case of weighted coverage (WC) functions was studied, using an approach that took a concave relaxation of the multilinear extension of such functions. Let $U$ be a set and $m : 2^U \to \mathbb{R}_+$ be a non-negative modular function. The ground set $V = \{B_1, B_2, \ldots, B_n\}$ is a collection of subsets of $U$. A weighted coverage function $f(S) : 2^V \to \mathbb{R}_+$ is defined as $f(S) = m(\cup_{B_i \in S} B_i)$. An equivalent formula is $f(S) = \sum_{u \in U} m(u) \min(1, |S \cap C_u|)$ where $C_u = \{B_i | u \in B_i\}$, which reveals that the weighted coverage function is actually a simple example of a one-layer DSF. In [20], Karimi et al. show that the multilinear extension of $f$ has a natural concave relaxation $\bar{F}(x) = \sum_{u \in U} m(u) \min(1, 1_{C_u} \cdot x)$ within a $1 - 1/e$ approximation. They first optimize the concave relaxation and claim that the solution is also good maximizer for the multilinear extension by the $1 - 1/e$ approximation. They further show that their approach yields solutions that match the $1 - 1/e$ guarantee of the continuous greedy algorithm, while reducing the computational cost by several orders of magnitude, mostly because they do not need to compute the multilinear extension.

Our framework in the present paper is a strict generalization of this previous method in the following ways: (1) The weighted coverage function class is a subclass of DSFs, and Karimi et al.'s proposed concave extension is a special case of a more general DSF concave extension; (2) We use a similar algorithmic approach which thus also has the advantage of better running time over the continuous greedy algorithm; and (3) We offer a still better bound for large $k$ where $k$ is the rank of the matroid.

As an example application, we note that DSFs generalize feature-based functions which are useful for various summarization tasks [21, 41, 17]. A feature-based function has the form $f(A) = \sum_{u \in U} w_u \phi_u(m_u(A))$ where $U$ is a set of features, $w_u > 0$ is a feature weight for $u \in U$, $m_u(A) =$

$\sum_{x \in X} m_u(A)$ is a feature-specific non-negative modular function, and $\phi_u(x)$ is a feature-specific monotone non-decreasing concave functions. Immediately, we have that the feature base functions are DSFs. Our proposed methods, therefore, offer a good bound for maximizing such functions if $\min_{u \in U} \frac{\min_{v \in V} m_u(v)}{\max_{v \in V} m_u(v)} k$ is large, which is fairly common in practice.

## 2 Background and Problem Setup

We assume every set function $f$ in this paper is normalized (i.e., $f(\emptyset) = 0$). A function $m$ is modular if and only if $m$ and $-m$ are both submodular. A normalized modular function $m(A)$ always has the form of $m(A) = \sum_{v \in V} m(v) = \mathbf{w} \cdot \mathbf{1}_A$, where $A \subseteq V$, $\mathbf{w}$ and $\mathbf{1}_A$ are $n$-dimensional vectors, $w = (m(1), m(2), \ldots, m(n))$ and $\mathbf{1}_A \in \mathbb{R}_+^V$ is 0 for coordinate $i \notin A$ and 1 for $i \in A$.

### 2.1 Matroid and matroid polytopes

A *matroid* $\mathcal{M} = (V, \mathcal{I})$ is a family of subsets of ground set $V$ with the following three properties:

1. $\emptyset \in \mathcal{I}$.
2. If $A \in \mathcal{I}$, then $B \in \mathcal{I}$ for all $B \subseteq A$.
3. For all $A, B \in \mathcal{I}$, if $|A| > |B|$, then there exists an element $v \in A \setminus B$, s.t. $B \cup \{v\} \in \mathcal{I}$.

The sets $I \in \mathcal{I}$ are the independent sets of the matroid. The third property ensures that the maximal independent sets always have the same size, equal to the rank $r_{\mathcal{M}} = k$ of the matroid. Matroids can be generalized to the continuous domain via the matroid polytope $\mathcal{P} = \mathrm{conv}(\mathbf{1}_A : A \in \mathcal{I})$ where "conv" means the convex hull.

### 2.2 Deep Submodular Function (DSFs)

A DSF [13, 3] $f$ is a natural generalization of feature-based functions and can be defined on a directed graph (Figure 2). The graph has $K + 1$ layers, where the first layer $V = V^{(0)}$ is the function's ground set, and additional layers $V^{(1)}, V^{(2)}, V^{(3)}, \ldots, V^{(K)}$ are sets of "features", "meta features", "meta-meta features", etc. The size of $V^{(i)}$ is $d^i = |V^{(i)}|$ for $i = 0, 1, 2, \ldots, K$. Note that the size of the final layer $V^{(K)}$ is always 1 because a DSF maps a

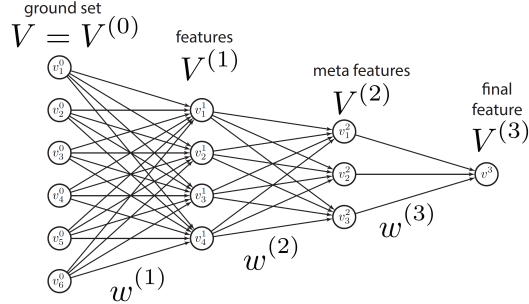

Figure 2: A layered DSF with $K = 3$ layers.

set to a real number. For any $i = 1, 2, \ldots, K$, two successive layers $V^{(i-1)}$ and $V^{(i)}$ are connected by a matrix $w^{(i)} \in \mathbb{R}_+^{d^i \times d^{i-1}}$. Therefore, matrix $w^{(i)}$ is indexed by $(v^i, v^{i-1})$ for $v^i \in V^{(i)}$ and $v^{i-1} \in V^{(i-1)}$. $w_{v^i}^{(i)}(v^{i-1})$ is an element from row $v^i$ and column $v^{i-1}$. We may think of $w_{v^i}^{(i)} : 2^{V^{(i-1)}} \to \mathbb{R}_+$ as a modular function defined on subset of $V^{(i-1)}$. Further, let $\phi_{v^i} : \mathbb{R}_+ \to \mathbb{R}_+$ be a non-negative, non-decreasing concave function. Thus, each element $v^i \in V^{(i)}$ has a modular function $w_{v_i}^{(i)}$ and concave function $\phi_{v^i}$ for $i = 1, 2, \ldots, K$. In this setting, a $K$-layer DSF $f : 2^V \to \mathbb{R}_+$ can be expressed, for any $A \subseteq V$, as follows:

$$f(A) = \bar{f}(A) + m_\pm(A), \qquad \text{where} \tag{1}$$

$$\bar{f}(A) = \phi_{v^K} \left( \sum_{v^{K-1} \in V^{(k-1)}} w_{v^K}^{(K)}(v^{k-1}) \phi_{v^{K-1}} \left( \cdots \sum_{v^2 \in V^{(2)}} w_{v^3}^{(3)}(v^2) \phi_{v^2} \left( \sum_{v^1 \in V^{(1)}} w_{v^2}^{(2)}(v^1) \phi_{v^1} \left( \sum_{a \in A} w_{v^1}^{(1)}(a) \right) \right) \right) \right). \tag{2}$$

#### 2.2.1 Concave functions $\phi_{v^i}$ and continuity

In a DSF, $\phi_{v^i}$ is a normalized (i.e., $\phi_{v_i}(0) = 0$) monotone non-decreasing concave function defined on $[0, +\infty)$. Via concavity, this implies that the function must also be continuous on $(0, +\infty)$. The only point that need not be continuous is $x = 0$, i.e., we may have $\lim_{x \to 0^+} \phi_{v^i}(x) > 0 = \phi_{v^i}(0)$. When used in a DSF, however, the set of possible input values to $\phi_{v^i}(x)$ is countable. Let $\beta > 0$ be the smallest strictly positive possible input to $\phi_{v^i}(x)$. We define another $\phi_{0,v^i} : \mathbb{R}_+ \to \mathbb{R}_+$ s.t.

$\phi_{0,v^i}(x) \equiv \phi_{v_i}(x)$ for $x \geq \beta$ and $\phi_{0,v^i}(x) \equiv \frac{\phi_{v_i}(\beta)}{\beta}x$ for $0 \leq x < \beta$. $\phi_{0,v^i}$ is normalized, monotone non-decreasing concave, and is continuous on $[0, +\infty)$. Moreover, replacing $\phi_{v^i}(x)$ with $\phi_{0,v^i}(x)$ leaves the DSF's valuation uncharged for any set. Therefore, w.l.o.g. we assume that all concave functions are also right-continuous at $x = 0$.

### 2.2.2 Final modular term $m_\pm$

Recall that $f(A) = \bar{f}(A) + m_\pm(A)$, where $\bar{f}(A)$ has the form of nested concave over modular and is always monotone non-decreasing, and $m_\pm(A)$ is a simple modular functions but can be negative. Although [13, 3] claim that the final modular function is sometimes useful in applications, this final function will change the optimization properties of $f$, since $\bar{f}$ is monotone non-decreasing but $f$ is non-monotone. In this work, we focus on the monotone non-decreasing DSF case where $m_\pm \geq 0$.[3]

### 2.3 DSF maximization

The problem we consider is DSF maximization, i.e.,

$$\text{Problem 1:} \quad \max_{A \in \mathcal{M}} f(A) \tag{3}$$

where $f$ is a DSF function and $\mathcal{M}$ is a matroid independence constraint. In this work, we focusing on solving this problem with the knowledge that $f$ is DSF.

## 3 Continuous extension of submodular functions

Although a submodular function is discrete, providing one value to each $A \subseteq V$, it is often useful to view such functions continuously. The bridge between the discrete and continuous worlds is made by a continuous extension of a submodular function, which is some function from the hypercube $[0,1]^n$ to $\mathbb{R}$ that agrees with $f$ on the hypercube vertices [14]. This includes the Lovász extension [29], which is the convex closure of the function, and also the multilinear extension [6], which is an approximation of the concave closure. In general, most continuous methods [6, 20] follow a similar strategy: they first find a continuous extension of $f$, then optimize it to obtain a fractional solution, and finally finish up by rounding the continuous solution back to a discrete final solution set.

In our framework, we use an extension that is tailor-made for a DSF.

### 3.1 A DSF's Natural Concave Extension

DSF functions have the form of nested sum of concave of modular (Equation (2)). [3] shows that there exists a natural concave extension of $f$ by replacing the discrete variables with real values in the nested form $F(x) = \bar{F}(x) + m_\pm \cdot x$ where

$$\bar{F}(x) = \phi_{vK}\left(\sum_{v^{K-1} \in V^{(k-1)}} w_{v^K}^{(K)}(v^{k-1})\phi_{v^{K-1}}\left(\cdots \sum_{v^2 \in V^{(2)}} w_{v^3}^{(3)}(v^2)\phi_{v^2}\left(\sum_{v^1 \in V^{(1)}} w_{v^2}^{(2)}(v^1)\phi_{v^1}\left(w_{v^1}^{(1)} \cdot x\right)\right)\right)\right). \tag{4}$$

Thus, $f(A)$ in Equation (1) has $f(A) = F(1_A)$ for all $A \subseteq V$. In fact, we have the following:

**Corollary 1** ([3])**.** *The DSF concave extension $F(x) : [0,1]^n \to \mathbb{R}$ is an extension of a DSF $f(x)$ and is concave.*

In [3], it is claimed that the extension is potentially useful for maximizing DSFs, possibly in a constrained fashion, followed by appropriate rounding methods, but the authors leave this as an open question. In the present work, we address this claim and answer this question in the affirmative. Before presenting our algorithm, we first discuss the relationship between DSF's natural concave extension and multilinear extension.

### 3.2 Multilinear extension

The concave closure of a submodular function $f$ is defined as $\min_{p \in \triangle^n(x)} \sum_{S \subseteq V} p_S f(S)$, where $\triangle^n(x) = \left\{ p \in \mathbb{R}^{2^n} : \sum_{S \subseteq V} p_S = 1,\ p_S \geq 0 \forall S \subseteq V,\ \&\ \sum_{S \subseteq V} p_S \mathbf{1}_S = x \right\}$. The concave closure is NP-hard even to evaluate [14]; hence, the multilinear extension is often used. We first specify the following definition:

**Definition 1.** *For a given $n$-dimensional vector $x \in [0,1]^n$, define $\mathcal{D}_x$ to be a distribution over sets $A$, s.t. $\Pr(A) = \Pi_{v \in A} x_v \Pi_{v \in V \setminus A}(1 - x_v)$.*

If we sample a random set $A$ from $\mathcal{D}_x$, then the event $v \in A$ is independent from $u \in A$ if $v \neq u$ and $\Pr(v \in A) = x_v$. With these definitions, we may define the multlinear extension as:

**Definition 2** (Multilinear Extension). $\mathcal{L}_f(x) = E_{A \sim \mathcal{D}_x} f(A)$.

Calinescu et al. [6] showed that we may solve the following continuous problem instead of solving Problem 1 directly.

$$\text{Problem 2:} \quad \max_{x \in \mathcal{P}} \mathcal{L}_f(x) \tag{5}$$

where $\mathcal{P}$ is a matroid polytope for $\mathcal{M}$.

Unfortunately, two problems remain with the multilinear extension $\mathcal{L}_f(x)$. First, calculating the exact value is not feasible in general, and even estimating it needs $O(n^5)$ time [6]. Second, it is **not** concave. Therefore finding the global maximizer of Problem 2 is in general not feasible. However, Calinescu et al. [6] developed a continuous greedy algorithm that finds $\hat{x}$ s.t. $\mathcal{L}_f(\hat{x}) \geq (1 - 1/e)\mathcal{L}_f(x^*)$ where $x^* \in \operatorname{argmax}_{x \in \mathcal{P}} \mathcal{L}_f(x)$. It is not hard to show that $\mathcal{L}_f(x^*) \geq f(A^*)$ where $A^* \in \operatorname{argmax}_{A \in \mathcal{M}} f(A)$, since $\mathcal{L}_f(1_{A^*}) = f(A^*)$. Therefore, $\mathcal{L}_f(\hat{x}) \geq (1 - 1/e)f(A^*)$. Next, we show how they round $\hat{x}$.

**Rounding** Rounding is a methodology that returns a discrete set from a fractional vector. "Pipage rounding" was first designed by Ageev et al. [2] and modified by Calinecu et al. [6] for submodular modular maximization, using a convex property of the multilinear extension. It maintains the quality of the solution in expectation, i.e., $E_{\hat{A} \sim \text{PIPAGE ROUNDING}(\hat{x})} f(\hat{A}) \geq \mathcal{L}_f(\hat{x})$, while satisfying the matroid constraint, thus finishing the proof sketch of the $1 - 1/e$ bounds for the continuous greedy algorithm. Another rounding technique is swap rounding [9] which can be seen as a replacement of pipage rounding with better running time $O(nk^2)$. In the special case of the matroid constraint, e.g., a simple partition matroid [8, 10][4], a simple rounding technique [5] is equivalent to pipage rounding with much easier implementation and linear running time. In our work, we can use any proper rounding techniques.

In this work, we show that given any DSF, it is not necessary to compute the multilinear extension at all. This is based on the following theorem:

**Theorem 1.** *For all $f \in$ DSF, its DSF concave extension $F$, and for all $x \in [0,1]^n$, we have $(1 - \delta)\left[1 - |V^{(1)}|e^{-\frac{\delta^2 \Delta(x)}{2}}\right] F(x) \leq \mathcal{L}_f(x) \leq F(x)$ where $\Delta(x) = \min_{v^1 \in V^{(1)}} \frac{w_{v^1}^{(1)} \cdot x}{\max_{v \in V} w_{v^1}(v)}$*

*Proof.* See Appendix A. $\qquad\square$

In Theorem 1, the term $\Delta(x)$ is fairly complex to interpret, but help can be gained by considering a lower bound of $\Delta(x)$ offered by the following lemma:

**Lemma 1.** $\Delta(x) \geq \frac{\|x\|_1 w_{\min}}{w_{\max}}$, *where* $w_{\max} = \max_{v^1 \in V^{(1)}} \max_{v \in V} w_{v^1}(v)$ *and* $w_{\min} = \min_{v^1 \in V^{(1)}} \min_{v \in V} w_{v^1}(v)$. *If $x$ is on the extreme point of a matroid polytope, then $\Delta(x) \geq \frac{k w_{\min}}{w_{\max}}$ where $k$ is the rank of the matroid.*

By applying Lemma 1 to Theorem 1 and $\Delta(x) = \Omega(k)$ and noticing $|V^{(1)}|e^{-\delta^2 \Omega(k)} \geq e^{-\delta^2 \Omega(k) + \log(|V^{(1)}|)} = e^{-\delta^2 \Omega(k)}$, we have the following results.

**Proposition 1.** $\max_\delta (1 - \delta)\left[1 - e^{-\delta^2 \Omega(k)}\right] F(x) \leq \mathcal{L}_f(x) \leq F(x)$

In Figure 1, we show that the coefficient of the lower bound converges to close to 1 as $k \to +\infty$.

Theorem 1 is one of the major results of the present work. It gives a concave relaxation (i.e., the natural concave extension of a DSF) of the non-concave multilinear extension $\mathcal{L}_f$. In this sense, we claim that multilinear extension $\mathcal{L}_f$ is closed to the DSF's natural concave extension $F$. Not surprisingly, maximizing a concave function is much easier than maximizing the multilinear extension for a variety of reasons.

**Lemma 2.** *Any concave problem solver that finds a solution $\hat{x}$ such that $F(\hat{x}) \geq (1-\epsilon)F(x_F^*)$ will satisfy $\mathcal{L}_f(\hat{x}) \geq (1-\epsilon)(1-\delta)\left[1-|V^{(1)}|e^{-\frac{\delta^2\Delta(\hat{x})}{2}}\right]\mathcal{L}(x_{\mathcal{L}}^*)$, where $x_F^*$ and $x_{\mathcal{L}}^*$ are the maximizer of the corresponding function subject to the matroid polytope membership.*

*Proof.* See Appendix B ☐

# 4 Projected Gradient Ascent

Following the general framework of [6, 20], we first find a fractional solution of the concave extension and then employ pipage rounding to obtain a feasible set. This approach offers the aforementioned guarantee for any member of the DSF family, regardless of its curvature.

## 4.1 Supergradient

For a concave function $F : \mathcal{P} \to \mathbb{R}$, where $\mathcal{P} \subseteq \mathbb{R}^n$ is a compact convex set, the set of supergradients of $f$ is defined as

$$\partial f(x) = \{g \in \mathbb{R}^n | f(y) - f(x) \leq g \cdot (y-x) \forall y \in \mathcal{P}\} \tag{6}$$

Given the formula of DSF concave extension $F(x)$, it is easy to compute supergradient as follows:

$$g(x)_e = \phi'_{v^K}(\cdot) \sum_{v^{K-1} \in V^{(k-1)}} \cdots \sum_{v^2 \in V^{(2)}} \sum_{v^1 \in V^{(1)}} \phi'_{v^{K-1}}(\cdot) \ldots \phi'_{v^2}(\cdot)\phi'_{v^1}(\cdot)w_{v^K}^{(K)}(v^{k-1}) \ldots w_{v^2}^{(2)}(v^1)w_{v^1}^{(1)}(e)$$

<div></div>

(7)

where $e \in [n]$ is a coordinate, $\phi'_{v^1}(\cdot)$ is the derivative of the concave function $\phi_{v^1}(x)$ at its current evaluation if it is differentiable, or is any supergradient of $\phi_{v^1}(x)$ if it is not differentiable. In fact, the way to calculate the supergradient of a DSF is exactly the same as what the backpropagation algorithm needs in deep neural network (DNN) training, and this was used in [13] to train DSFs. This is also one of the reasons for the

---
**Algorithm 1:** Projected Gradient Ascent [4]

---
**input** : DSF concave extension $F$, matroid polytope $\mathcal{P}$, learning rate $\eta$, maximum number of iterations $T$

Let $x^{(0)} \leftarrow \operatorname{argmin}_{x \in \mathcal{P}} \|x\|_2^2$

**for** $t = 1, 2, \ldots, T$ **do**

    compute a supergradient $g(x^{(t-1)}) \in \partial F(x^{(t-1)})$ ;

    $x^{(t)} \leftarrow \operatorname{argmin}_{x \in \mathcal{P}} \left\|x - \left(x^{(t-1)} + \eta g(x^{(t-1)})\right)\right\|_2^2$;

    // This is done by projecting $x^{(t-1)} + \eta g(x^{(t-1)})$ to $\mathcal{P}$

**end**

**return** $\frac{1}{T}\sum_{t=1}^{T} x^{(t)}$

---

name *deep submodular functions*. Therefore, all of the toolkits available for DNN training, with provisions for automatic symbolic differentiation (e.g., PyTorch [33] and TensorFlow [1] ) can be used to maximize a DSF. Since they are optimized for fast GPU computing, they can offer great practical and computational advantages over traditional submodular maximization procedures.

## 4.2 Projected gradient Ascent

We utilize the following theorem from [4, 7] (modified for the concave, rather than convex, case) to establish our bounds for DSF-based submodular maximization.

**Theorem 2.** *[[4, 7]] For any concave function $F : \mathbb{R}_+^n \to \mathbb{R}$, let $R^2 = \sup_{x \in \mathcal{P}} \|x\|_2^2$ and $B^2 = \sup_{x \in \mathcal{P}} \|g(x)\|_2^2$, Algorithm 1 with learning rate $\eta = \sqrt{\frac{R}{BT}}$ will obtain a fractional solution $\hat{x}$ s.t. $F(\hat{x}) \geq \max_{x \in \mathcal{P}} F(x) - RB\sqrt{\frac{2}{T}}$.*

Applying Theorem 2 to Algorithm 1 and using our propose concave function $F(x)$, we have the following result:

**Lemma 3.** *For any $0 < \epsilon < 1$, Algorithm 1 will obtain a fractional $\hat{x}$ s.t. $f(\hat{x}) \geq (1-\epsilon)\max_{x \in \mathcal{P}} f(x)$ with running time $T = O(n^2\epsilon^{-2})$.*

*Proof.* See Appendix C. ☐

Thus, we have a approximate solution to the concave maximization problem and using this, in concert with Lemma 2, we arrive at the following which offers a guarantee of our proposed method.

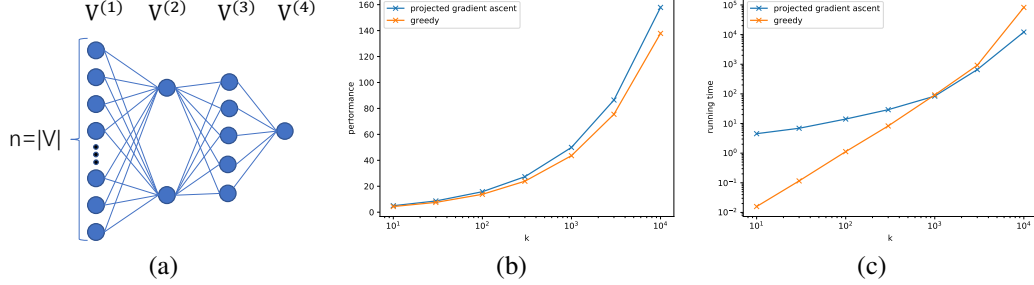

$$V^{(1)} \quad V^{(2)} \quad V^{(3)} \quad V^{(4)}$$

n=|V|

(a)          (b)          (c)

Figure 3: (a) DSF structure (b) performance comparison, solution value vs. $k$, (c) running time vs. $k$.

**Theorem 3.** *Algorithm 1 with pipage rounding will give $\hat{X}$ such that $Ef(\hat{X}) \geq \max_{0<\delta<1}(1 - \epsilon)(1 - \delta)\left[1 - |V^{(1)}|e^{-\frac{\delta^2 w_{\min} k}{w_{\max}}}\right] \max_{X \subseteq \mathcal{M}} f(X)$ with running time $T = O(n^2 \epsilon^{-2})$*

In Figure 1, we have a comparison of this bound with the traditional $1/2$ and $1 - 1/e$ bounds. We find our proposed bound approaches 1 when $k \to +\infty$ and beats other bounds for large $k$ ($> 10^4 \sim 10^6$, depending on $w_{\min}/w_{\max}$).

**Corollary 2.** *Algorithm 1 with with pipage rounding will give $\hat{X}$ such that $Ef(\hat{X}) \geq \max_{0<\delta<1}(1 - \epsilon - \delta - e^{-\delta^2 \Omega(k)}) \max_{X \subseteq \mathcal{M}} f(X)$ with running time $T = O(n^2 \epsilon^{-2})$*

## 5 Experiments

In this section, we perform a number of synthetic dataset experiments in order to demonstrate proof of concept and also to offer empirical evidence supporting our bounds above. While the results of the paper are primarily theoretical, the results of this section show that our methods can yield practical benefit and also demonstrate the potential of the above methods for large-scale DSF-constrained maximization.

Figure 3 shows the structure of the DSF $f : 2^V \to \mathbb{R}_+$ to be maximized. It is a three-layer DSF having ground set $V = V^{(0)}$ with $|V| = n$. We partition the ground set $V$ into blocks $V_1 \cup V_2 \cup V_3$ s.t. $|V_1| = |V_2| = |V_3| = t$, where $t = |V|/3$. In the next layer $V^{(1)}$, the inner part of $f$ consists of two concave-composed-with-modular functions, $f_{1,1}(A) = \min(|X \cap [V_1 \cup V_3]| + \alpha|X \cap V_2|, t)$ and $f_{1,2}(A) = \alpha|X \cap [V_1 \cup V_3]| + |X \cap V_2|$ where $\alpha = 0.1$ is a parameter. In the subsequent layer $V^{(2)}$, every node is concave over the weighted sum of $f_{1,1}(A)$ and $f_{1,2}(A)$, i.e., $f_{2,i}(A) = \sqrt{w_i^{(2)}(1)f_{1,1}(A) + w_i^{(2)}(2)f_{1,2}(A)}$ for $i \in V^{(2)}$, where $w_i^{(2)}$ is a 2-dimensional uniformly at random vector from $[0, 1]^2$. Finally, for the last layer $V^{(3)}$, the entire function $f(A) = \sum_{i \in V^{(2)}} w^{(3)}(i)f_{2,i}(A)$, where $w^{(3)}$ is a $|V^{(2)}|$−dimensional vector again uniformly at random from $[0, 1]$. The matroid constraint is a partition matroid s.t. $X$ is independent if $|X \cap \{v_{1,i}, v_{2,i}\}| \leq 1$ for $i = 1, 2, \ldots, t$, where we label $V_1 = \{v_{1,i}\}_{i=1}^t$ and $V_1 = \{v_{2,i}\}_{i=1}^t$. The rank of this matroid is therefore $k = 2t$. We repeat the experiment on 30 random DSFs. For each DSF, we maximize it by each algorithm and take the average function value, respectively. Figure 3(b) shows the performance of our method compared to the combinatorial greedy algorithm using the lazy evaluation trick [30]. We see that our method offers a solution that is consistently better than the standard greedy for all $k$. Regarding running time, we find that while our method is slower than lazy greedy for small $k$, it becomes faster than lazy greedy for large $k$ (Figure 3c). For a fair comparison, both algorithms were implemented in Python and run on a single CPU. We anticipate that our method will run even faster on parallel GPU machines, which can be accomplished easily using any modern DNN toolkit (e.g., PyTorch [33] or TensorFlow [1]).

Acknowledgments: This material is based upon work supported by the National Science Foundation under Grant No. IIS-1162606, the National Institutes of Health under award R01GM103544, and by a Google, a Microsoft, and an Intel research award. This research is also supported by the CONIX Research Center, one of six centers in JUMP, a Semiconductor Research Corporation (SRC) program sponsored by DARPA.

## Footnotes

[1]A submodular function is said to be monotone non-decreasing if $f(v|A) \geq 0$ for all $v \in V$ and $A \subseteq V$.

[2]Multilinear extension is a special case of continuous DR-submodular.

[3]Note, if $m_\pm$ is non-negative, it can merge into $\bar{f}(A)$ which is equivalent to $m_\pm = 0$

[4] $\mathcal{I} = \{A \subseteq V \mid |A \cup V_i| \leq 1 \; \forall i\}$ where $\{V_i\}$s are a partition of $V$.

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
