[Supplementary Material · dsf_max_camera_ready_supplementary.pdf]

# A   Proof of Theorem 1

Before going to the proof, we discuss the following (adopted) lemma,

**Lemma 4** (Adopted from [34])**.** *Let $a_1, a_2, \ldots, a_r$ be reals in (0,1]. Let $X_1, X_2, \ldots, X_n$ be independent Bernoulli trails with $E[X_j] = p_j$. For the random variable $\Psi = \sum_{j=1}^{n} a_j X_j$ and for $\delta \in (0, 1]$,*

$$\Pr[\Psi < (1 - \delta)E\Psi] < e^{\frac{-\delta^2 E\psi}{2}} \tag{8}$$

This lemma is particularly interesting since the multilinear extension $\mathcal{L}_f(x) = E_{A \sim D_x} f(A)$ is a result of sampling. The relationship is more clear when we consider modular $m(A) = w \cdot 1_A$. Although, it is easy to show that $\mathcal{L}_m(x) = w \cdot x$, we can think it differently. Ignore for the moment $\mathcal{L}_m(x) = w \cdot x$. By definition, $\mathcal{L}_m(x) = E_{A \sim D_x} m(A) = E_{A \sim D_x} w \cdot 1_A$ and $1_A$ is actually independent Bernoulli trails. From lemma 4, we know there is a high probability of having "good" $1_A$ from $A \sim D_x$. By "good", we mean the following:

**Definition 3.** *In the context of $A \sim D_x$, for any DSF $f$, let the event $B_f(\delta)$ be $f(A) \geq (1 - \delta)F(x)$, where $F$ is the DSF concave extension of $f$. Let $B(\delta) = \wedge_{v \in V_1} B_{m_{v_1}}(\delta)$.*

When calculating the lower bound of $\mathcal{L}_m(x)$, we simply throw away all "bad" instance and still, the summation of "good" instance can be high, i.e., $\mathcal{L}_m(x) \geq (1 - \delta) \Pr(B_m(\delta))F(x)$, where $F(x) = w \cdot x$ is the DSF concave extension of $m$.

Now we begin to prove theorem 1 by generalizing the above discussion to DSFs.

**Theorem 1.** *For all $f \in$ DSF, its DSF concave extension $F$, and for all $x \in [0, 1]^n$, we have*
$$(1 - \delta) \left[ 1 - |V^{(1)}| e^{-\frac{\delta^2 \Delta(x)}{2}} \right] F(x) \leq \mathcal{L}_f(x) \leq F(x) \text{ where } \Delta(x) = \min_{v^1 \in V^{(1)}} \frac{w_{v^1}^{(1)} \cdot x}{\max_{v \in V} w_{v^1}(v)}$$

First we show the upper bound of the multilinear extension. By definition, $F(x)$ is an extension of $f(x)$ so they agree on the integer values, which means for all $A \sim \mathcal{D}_x$, $f(A) = F(1_A)$. Therefore, $\mathcal{L}_f = E_{A \sim \mathcal{D}_x} f(A) = E_{A \sim \mathcal{D}_x} F(1_A)$. On the other hand, $F(x) = F(E_{A \sim \mathcal{D}_x} 1_A)$ and $F(x)$ is concave. By Jensen's inequality, we have $\mathcal{L}_f = E_{A \sim \mathcal{D}_x} F(1_A) \leq F(E_{A \sim \mathcal{D}_x} 1_A) = F(x)$.

The proof of lower bound is more complex. We notice that, given the definition, a DSF has the form of multiple concave over a sum of modular functions, as stated in Equation (2). Let $m_{v_1}(A) = \sum_{a \in A} w_{v_1}^{(1)}(a) = w_{v_1}^{(1)} \cdot 1_A$. Immediately, we see that $f(A)$ is depends on the values of $m_{v_1}(A)$ for all $v_1 \in V^{(1)}$

$$f(A) = \phi_{v^K} \left( \sum_{v^{K-1} \in V^{(k-1)}} w_{v^K}^{(K)}(v^{k-1}) \phi_{v^{K-1}} \left( \ldots \sum_{v^2 \in V^{(2)}} w_{v^3}^{(3)}(v^2) \phi_{v^2} \left( \sum_{v^1 \in V^{(1)}} w_{v^2}^{(2)}(v_1) \phi_{v^1} \left( m_{v_1}(A) \right) \right) \right) \right) \tag{9}$$

We know that $\mathcal{L}_f = E_{A \sim \mathcal{D}_x} f(A)$ and, ultimately, we want to show that $\mathcal{L}_f \geq C \cdot F(x)$ for some $C > 0$. The intuition is that if we could calculate $\Pr[f(A) \geq C_1 F(x)]$, then we will have $\mathcal{L}_f \geq \Pr[f(A) \geq C_1 F(x)] \cdot C_1 \cdot F(x)$. Like the discussion at the beginning of this section, we began by analyzing modular functions $m_{v_1}(A)$ for $v_1 \in V_1$. Notice that in $m_{v_1}(A) = w_{v_1}^{(1)} \cdot 1_A$, coordinates of $1_A$ are independent Bernoulli trails and applying Lemma 4 to $m_{v_1}(A)$, we claim that for $v_1 \in V_1$ and for $A \sim D_x$,

$$\Pr\left[ m_{v_1}(A) < (1 - \delta)E_{A \sim D_x} m_{v_1}(A) \right] < e^{-\frac{\delta^2 E_{A \sim D_x} m_{v_1}(A)}{2 \max(w_{v_1}^{(1)})}} \tag{10}$$

$$\Pr\left[ m_{v_1}(A) < (1 - \delta) w_{v_1}^{(1)} \cdot x \right] < e^{-\frac{\delta^2 w_{v_1}^{(1)} \cdot x}{2 \max(w_{v_1}^{(1)})}} \tag{11}$$

$$\Pr\left[ \bar{B}_{m_{v_1}}(\delta) \right] < e^{-\frac{\delta^2 w_{v_1}^{(1)} \cdot x}{2 \max(w_{v_1}^{(1)})}} \tag{12}$$

since the DSF concave extension of a modular function is the same as the multilinear extension.

Note that $\{B_{v_1}\}$s may or may not be independent events, but the following is always true,

$$\Pr\left[\bar{B}(\delta)\right] \leq \sum_{v_1 \in V_1} \Pr\left[\bar{B}_{m_{v_1}}(\delta)\right] \tag{13}$$

where $B(\delta) = \wedge_{v \in V_1} B_{m_{v_1}}(\delta)$. Then we show the properties $B_f(\delta)$ are preserved by weighted sums and concave functions.

**Lemma 5.** *When sampling $A \sim D_x$, for any two DSF $f_1$ and $f_2$, let $f = f_1 + f_2$, we have $B_{f_1}(\delta) \wedge B_{f_2}(\delta) \to B_f(\delta)$.*

*Proof.* When $B_{f_1}(\delta) \wedge B_{f_2}(\delta)$ is **True**, we have $f_1(A) \geq (1-\delta)F_1(A)$ and $f_2(A) \geq (1-\delta)F_2(A)$. Notice that $F(A) = F_1(A) + F_2(A)$, so immediately, we have $f(A) \geq (1-\delta)F(A)$. $\square$

**Lemma 6.** *When sampling $A \sim D_x$, for monotone non-decreasing concave function $\phi$ and DSF $f$, we have $B_f(\delta) \to B_{\phi(f)}(\delta)$.*

*Proof.* Immediately, we notice that the DSF concave extension of $\phi(f(A))$ is $\phi(F(x))$.

Looking at the curve of $\phi(x)$, if we connect $(0,0)$ and $(y, \phi(y))$ with a line, then the curve is above it for $x \in [0, y]$. Therefore $\frac{\phi(x)}{x}$ is a decreasing function.

If $B_f(\delta)$ is **True**, we have $f(A) \geq (1-\delta)F(x)$. then $\frac{\phi(f(A))}{\phi(F(x))} \geq (1-\delta)\frac{\phi((1-\delta)F(x))}{(1-\delta)\phi(F(x))} \geq (1-\delta)\frac{\phi(F(x))}{\phi(F(x))} = 1 - \delta$. Therefore $B_{\phi(f)}(\delta)$ is **True**.

$\square$

As the basic structure of DSF $f$, we can get $f$ by recursively applying concave functions or weighted sums over $m_{v_1}$ [3]. Therefore, applying Lemmas 5 and 6, we claim that, when sampling $A \sim D_x$, if $B(\delta)$ is **True** then $B_f(\delta)$ is **True**, which means

$$\Pr\left[B_f(\delta)\right] \geq \Pr\left[B(\delta)\right] \tag{14}$$

$$\geq 1 - \sum_{v_1 \in V_1} \Pr\left[\bar{B}_{m_{v_1}}(\delta)\right] \tag{15}$$

$$\geq 1 - \sum_{v_1 \in V_1} e^{-\frac{\delta^2 w_{v_1}^{(1)} \cdot x}{2 \max(w_{v_1}^{(1)})}} \tag{16}$$

Therefore,

$$\mathcal{L}_f(x) = E_{A \sim \mathcal{D}_x} f(A) \tag{17}$$
$$\geq (1-\delta)\Pr\left[B_f(\delta)\right] F(x) \tag{18}$$

$$\geq (1-\delta)\left[1 - \sum_{v_1 \in V_1} e^{-\frac{\delta^2 w_{v_1}^{(1)} \cdot x}{2 \max(w_{v_1}^{(1)})}}\right] F(x) \tag{19}$$

since $f(A)$ is always non-negative, thus finished the proof of the lower bound of $\mathcal{L}$.

# B   Proof of Lemma 2

**Lemma 2.** *Any concave problem solver that finds a solution $\hat{x}$ such that $F(\hat{x}) \geq (1-\epsilon)F(x_F^*)$ will satisfy $\mathcal{L}_f(\hat{x}) \geq (1-\epsilon)(1-\delta)\left[1 - |V^{(1)}|e^{-\frac{\delta^2 \Delta(\hat{x})}{2}}\right] \mathcal{L}(x_{\mathcal{L}}^*)$, where $x_F^*$ and $x_{\mathcal{L}}^*$ are the maximizer of the corresponding function subject to the matroid polytope membership.*

*Proof.* Given Theorem 1, we have $\mathcal{L}_f(\hat{x}) \geq (1-\delta)\left[1 - |V^{(1)}|e^{-\frac{\delta^2 \Delta(\hat{x})}{2}}\right] F(\hat{x}) \geq (1-\delta)\left[1 - |V^{(1)}|e^{-\frac{\delta^2 \Delta(\hat{x})}{2}}\right] F(\hat{x}) \geq (1-\epsilon)(1-\delta)\left[1 - |V^{(1)}|e^{-\frac{\delta^2 \Delta(\hat{x})}{2}}\right] F(x_{\mathcal{L}}^*) \geq (1-\epsilon)(1-\delta)\left[1 - |V^{(1)}|e^{-\frac{\delta^2 \Delta(\hat{x})}{2}}\right] \mathcal{L}_f(x_{\mathcal{L}}^*)$ $\square$

## C   Proof of Lemma 3

**Lemma 3.** *For any $0 < \epsilon < 1$, Algorithm 1 will obtain a fractional $\hat{x}$ s.t. $f(\hat{x}) \geq (1 - \epsilon) \max_{x \in \mathcal{P}} f(x)$ with running time $T = O(n^2 \epsilon^{-2})$.*

*Proof.* Given Theorem 2, we have

$$f(\hat{x}) \geq \max_{x \in \mathcal{P}} f(x) - RB\sqrt{\frac{2}{T}} \tag{20}$$

$$\geq \left(1 - \frac{RB}{\max_{x \in \mathcal{P}} f(x)}\sqrt{\frac{2}{T}}\right) \max_{x \in \mathcal{P}} f(x) \tag{21}$$

Let $\epsilon = \frac{RB}{f(x^*)}\sqrt{\frac{2}{T}}$ and $x^* \in \operatorname{argmax}_{x \in \mathcal{P}} f(x)$. Therefore, $T = \frac{2R^2 B^2}{\epsilon^2 f^2(x^*)}$.

We have $R^2 = \sup_{x \in \mathcal{P}} \|x\|_2^2 \leq n$. $B^2 = \sup_{x \in \mathcal{P}} \|g(x)\|_2^2$ and $g(x)$ is a supergradient of $f(x)$ and the maximizer of $\|g(x)\|$ is $x = \mathbf{0}$ since $f(x)$ is concave. Note than $g(0)_e$ is only related to the parent nodes of $e$,

$$g(0)_e = \phi'_{v^K}(0) \sum_{v^{K-1} \in V^{(k-1)}} \cdots \sum_{v^2 \in V^{(2)}} \sum_{v^1 \in V^{(1)}} \phi'_{v^{K-1}}(0) \dots \phi'_{v^2}(0)\phi'_{v^1}(0) w_{v^K}^{(K)}(v^{k-1}) \dots w_{v^2}^{(2)}(v^1) w_{v^1}^{(1)}(e) \tag{22}$$

Therefore $\left\|\frac{g(0)_{e_1}}{g(0)_{e_2}}\right\| \leq \frac{w_{\max}}{w_{\min}}$ for all coordinates $e_1$ and $e_2$. So $B^2 = \sup_{x \in \mathcal{P}} \|g(x)\|_2^2 \leq \frac{w_{\max}}{w_{\min}} ng(0)_e = O(n)$. And for the last unknown term in $T$, $f(x^*)$ is not decreasing with respect to $n$. So we thus have that $T \leq O(n^2 \epsilon^{-2})$.

$\square$

## D   Cases for Fully Curved DSFs

The current best curvature bound for submodular maximization under a matroid constraint is $1 - {}^c\!/_e$ [37] where $c$ is the curvature of submodular $f$. We claim our bound (Theorem 3) is **not** curvature related and have shown that it is better than $1 - {}^1\!/_e$ for large $k$ in figure 1. But the readers may suspect that there might be a hidden dependency on curvature, and the case that our bound beats $1 - {}^1\!/_e$ is all for low curvature, where traditional methods also have near optimal performance. In response, we show by the following lemma which states that our proposed guarantee can be maximized even for fully curvature DSF functions.

**Lemma 7** (Bound is maximized even if fully curved). *There exists a fully curved [5] DSF $f(A)$, and Algorithm 1 with pipage rounding will give $\hat{X}$ such that $Ef(\hat{X}) \geq \max_{0 < \delta < 1}(1 - \epsilon)(1 - \delta)\left[1 - e^{-\delta^2 k}\right] \max_{X \subseteq \mathcal{M}} f(X)$ with running time $T = O(n^2 \epsilon^{-2})$*

*Proof.* The example is fairly simple, $f(A) = \min(|A|, a)$ where $1 \leq a \leq |A| - 1$ is a constant number and the matroid constraint is $|A| \leq k$. Immediately, we notice that $c \equiv 1 - \min_{v \in V} \frac{f(v|V\setminus\{v\})}{f(v)} = 1$ so that $f$ is fully curved. Even though this is a very easy optimization problem and all algorithms can find the optimal solution, [37] still classifies it as the hardest case by curvature and their bound is at its worst $1 - {}^1\!/_e$.

On the other hand, according to Theorem 3, our proposed bound is at its best as $\max_{0 < \delta < 1}(1 - \epsilon)(1 - \delta)\left[1 - e^{-\delta^2 k}\right]$ and not affected by curvature. It is easy to find out that our bound surpasses $1 - {}^1\!/_e$ for $k \geq 54$ if we set $\epsilon = 0.01$ and for $k \geq 88$ if $\epsilon = 0.1$.

$\square$

**Theorem 1.** *Algorithm 1 with pipage rounding will give $\hat{X}$ such that $Ef(\hat{X}) \geq \max_{0 < \delta < 1}(1 - \epsilon)(1 - \delta)\left[1 - |V^{(1)}|e^{-\frac{\delta^2 w_{\min} k}{w_{\max}}}\right] \max_{X \subseteq \mathcal{M}} f(X)$ with running time $T = O(n^2 \epsilon^{-2})$*

# E Algorithm for pipage rounding

Algorithm 3 is the main procedure. During each iteration, a subroutine, Algorithm 2 HITCON-STRAINT$(y, i, j)$, is called to find the nearest constraint in the direction of $e_i - e_j$. It returns the intersection and a smaller tight set[6] containing $i$ [6].

---

**Algorithm 2:** HITCONSTRAINT [5]

**input** : $y, i, j$
Let $\mathcal{A} \leftarrow \{A \subseteq V | i \in A, j \notin A\}$;
Let $\delta \leftarrow \min_{A \in \mathcal{A}}(r_{\mathcal{M}}(A) - y(A))$ and $A \in \operatorname{argmin}_{A \in \mathcal{A}}(r_{\mathcal{M}}(A) - y(A))$
**if** $y_j < \delta$ **then**
$\quad | \quad y_i \leftarrow y_j - \delta, y_j \leftarrow 0, A' \leftarrow A$;
**else**
$\quad | \quad y_i \leftarrow y_i + \delta, y_j \leftarrow y_j - \delta, A' \leftarrow A$;
**return** $(y, A')$

---

**Algorithm 3:** Pipage rounding [5]

**input** : matroid $\mathcal{M}$, fractional $x$
**while** *x is not integral* **do**
$\quad | \quad T \leftarrow X$;
$\quad | \quad$ **while** *T contains fractional variables* **do**
$\quad | \quad | \quad$ Pick $i, j \in T$ fractional;
$\quad | \quad | \quad (y^+, A^+) \leftarrow$ HITCONSTRAINT$(y, i, j)$;
$\quad | \quad | \quad (y^-, A^-) \leftarrow$ HITCONSTRAINT$(y, j, i)$;
$\quad | \quad | \quad$ **if** $y^+ = y^- = y$ **then**
$\quad | \quad | \quad | \quad T \leftarrow A^+$;
$\quad | \quad | \quad$ **else**
$\quad | \quad | \quad | \quad p \leftarrow \|y^+ - y\| / \|y^+ - y^-\|$;
$\quad | \quad | \quad | \quad$ With probability p, $\{y \leftarrow y^-, T \leftarrow T \cap A^-\}$;
$\quad | \quad | \quad | \quad$ Else $\{y \leftarrow y^+, T \leftarrow T \cap A^+\}$;
$\quad | \quad$ **end**
**end**
**return** $\frac{1}{T} \sum_{t=1}^{T} x^{(t)}$

## Footnotes

[5] $c \equiv 1 - \min_{v \in V} \frac{f(v|V\setminus\{v\})}{f(v)} = 1$

[6]For $y \in \mathcal{P}(\mathcal{M})$, a set $A \subseteq V$ is tight if the rank of $A$ in $\mathcal{M}$ equals $y(A)$.