[Reviews · NeurIPS 2018]

Reviewer 1



The paper proves a very interesting result: For maximizing Deep Submodular Functions (DSF) with matroid constraints, one can provide efficient algorithms that, under mild assumptions on the singleton marginals, have approximation factor better than 1-1/e (and potentially approaching 1 when the rank of the matroid becomes large). This is given in Theorem 1 which I think is the main result of the paper. The basic idea behind the algorithm is that for DSFs there is a natural concave extension, equations (4), (5) that can be maximized by projected gradient ascent (this result has been proved in [3]). The authors show in Proposition 1 that this concave extension is close to the multilinear extension, and in section 4 they show that the projected gradient ascent algorithm can be implemented efficiently (e.g. the subgradient of the concave extension can be computed efficiently due to the deep structure). The paper is written well and the results are novel. However, I have a few comments for which I would appreciate a response from the authors: 1) How is the projection step in Algorithm 1 done? E.g. what is the complexity for general matroids? Since you prefer k being large, it would be useful to provide the exact runtime of the algorithm in terms of n and k. Also, in Figure 3 (c), is the projection step considered? (I guess for partition matroid the projection step is easier than general matroids) 2) Can the authors explain how the bounds of Theorem 1 compare with 1-1/e if k is not so large? (lets assume that the modular term m_{+-} in equation (1) is zero to make the overall objective monotone). Also, it would be beneficial if the authors could explain how the bound in Theorem 1 would be in the worst case. I.e. given a fixed k and perhaps negative modular terms, how bad the bounds of Theorem 1 could be? In particular, are they better than 1/2? 3) In your experiments, I think it would be useful in include settings with negative modular terms (m_{+-} in equation (1)). Some minor comments: - Line 40: befit -> benefit - Regarding the discussion in lines 74-78, there is now an efficient and practical algorithm for maximizing the multilinear extension over matroids (see the stochastic continuous greedy algorithm by Mokhtari et al, 2017). Basically, you don't need to compute the multilinear extension at each step (unlike [6]) and only an unbiased estimator (that can be computed very efficiently) is enough to maximize the function using stochastic optimization algorithms.

Reviewer 2



Summary: Deep submodular functions are a class of submodular functions that may be modeled using a neural network. This work focuses on maximising deep submodular functions with matroid constraints. This is done using multilinear extension that can be optimized using gradient asnet. Further, pipage rounding is used to find an approximate optimal solution to the original problem. Pros: - It is first novel attempt to maximize deep submodular funcions under matroid constraints. - The paper is well structured and well written - Theorem 1 where they prove that multilinear extensive of DSF need not be calculated to optimize the function is the major result - Theorem 3 gives an approximation due to pipage rounding negatives/clarifications: One of the main drawbacks of the paper is experimental evaluation. One of the issue with deep neural networks is the amount it takes to optimize the function in the parameter. I believe, the goal of deep submodular functions is to leverage the underlying function structure to be able to speed up the optimize algorithms yet as express as DNNs. However, this paper does not demosnstrate this. Update after authors response: I do agree with the authors that NIPS should have space for theoretical contribution and that is precisely the reason why I tend to vote towards accepting and upgrading to 7. However, I believe, a good experimental evaluation would have further strengthened the paper.

Reviewer 3



This paper studies gradient-based methods for optimizing a deep submodular function. The starting point is a natural concave relaxation where each variable is allowed to take fractional values. The main contribution is a bound on the gap between the concave relaxation and the multilinear extension, generalizing a bound for coverage functions by Karimi et al. For large k, and depending on the parameters of the DSF, the resulting approximation ratio can improve on 1 - 1/e. Providing improved tools for maximizing DSFs is a natural problem, and the ability to improve on the approximation ratio and scalability of the greedy algorithm is an attractive property. However, the improvements (both theoretical and empirical) appear somewhat marginal in many cases. Additional information on a few points would be helpful to assess the significance of the work: First, the bounds depend on parameters of the DSF like |V_1| and w_min/max. It is claimed that these quantities are often favorable in practice. However, none of the papers cited use DSFs. What evidence is there that real problem instances are amenable to such techniques? Second, what is the value of n (the size of the ground set) for the experiments? This is important context for understanding the scalability comparison vs greedy. Third, the comparison vs greedy uses a partition matroid constraint. Is there any improvement in solution quality (compared to greedy) for a uniform matroid? This is the more common case in practice, and would help disentangle two factors that may be driving the performance improvement: (a) the concave relaxation becoming tighter with large k, and (b) continuous methods in general simply being more appropriate for general matroids. Fourth, the motivation for using the concave relaxation is that the multilinear extension might require many samples to approximate. However, this is often pessimistic because the multilinear extensions of common families such as converage functions or facility location can be computed in closed form. Do similar techniques extend to the compositions used in DSFs? Update after author response: The response helps address some of my concerns and I have raised my score by one point. The experimental evaluation is still weak overall though; this paper introduces some nice ideas and it would be good to see them tested more thoroughly.

Reviewer 4



The submodular maximization is an important problem with many real-world applications. One powerful way of tackling these problems is to consider their contentious relaxations, then maximize those relaxations, and finally rounding the resulting solutions into discrete answers. In this paper, the authors consider maximizing a special class of submodular functions called deep submodular functions (DSFs) under a matroid constraint. The (almost) standard approach to solve a submodular maximization problem in the continuous domain is to consider the multilinear extension of the original set function. The contribution of this paper is to show that the natural concave extension of a DSF function provides an acceptable approximation for the multilinear extension. Because this natural extension is concave, a gradient ascent method will find the maximum value. The idea of bounding multilinear extension by a concave function for coverage functions is already introduced in [1]. In this paper, authors provide a tighter bound for a more general class of functions. I have read the paper in details and I believe it is technically correct. Next you can find my detailed comments. 1. This works is along the new progresses on maximizing DR-continuous functions (and in particular multi-linear extension). For example, Hassani et al. [2] recently have proven that stochastic gradient ascent achieves 1/2 approximation for DR-continuous functions. Furthermore, Mokhtari et al. [3] have developed practical algorithms for stochastic conditional gradient methods which mitigates the need for subsampling while achieves the (1-1/e)-approximation guarantee. The computational complexity of the algorithm from [3] is much better than that of O(n^7). The references (and comparison) to these closely related papers are missing. 2. The bound in Theorem 3 is a function of k, |V^1|, w_min and w_max. DSFs are a narrow family of submodular functions. The dependence on |V^1|, w_min and w_max might makes the coverage of these DSFs even narrower (for the cases their algorithm provides improvements for the approximation guarantee). For example, it might be the case that |V^1| should be very large in order to cover a broad family of submodular functions. I believe that the authors should have elaborated (more) on the dependence on these parameters. Considering these points and the fact that the result is only for a subclass of submodular functions with a matroid constraint, I find the contribution of this paper somewhat marginal. 3. The big O notation is not used correctly. Note that O(k) could be 0, and the approximation guarantee could be arbitrarily bad. Also, it is not mathematically sound to write \Delta(x) \geq O(k) (line 243). 4. The Experiments section of the paper is weak as it only considers synthetic datasets. I think adding experiments over the real world datasets would improve the quality of this paper. 5. Minor comments: a) The presentation of paper could be improved. Next you can find several suggestions: In page 1, "A" is not defined. Figure 2 is discusses before Figure 1. Font of equations (2) and (5) is very small. It is better to write the equation (2) in an iterative formulation. Different notations are used for definitions: \equiv and \triangleq. The fractions in the text are sometimes written by \nicefrac{}{} and sometimes \frac{}{}. Some sentences starts with numbers, e.g., line 80. The punctuation for "i.e.," and "e.g.," is not consistent. b) line 40: benefit c) line 75: \epsilon^{-2} d) line 178: "we are focusing" or "we focus" e) line: it is better to say "in continuous domain" f) line 204: the definition of concave closure seems unnecessary g) line 289: proposed References: [1] Mohammad Reza Karimi, Mario Lucic, S. Hamed Hassani, Andreas Krause. Stochastic Submodular Maximization: The Case of Coverage Functions. NIPS 2017. [2] S. Hamed Hassani, Mahdi Soltanolkotabi, Amin Karbasi. Gradient Methods for Submodular Maximization. NIPS 2017. [3] Aryan Mokhtari, Hamed Hassani, Amin Karbasi. Conditional Gradient Method for Stochastic Submodular Maximization: Closing the Gap. AISTATS 2018.